# A Nomenclatural and Taxonomic Revision of the *Senecio squalidus* Group (Asteraceae)

**DOI:** 10.3390/plants11192597

**Published:** 2022-10-02

**Authors:** Giulio Barone, Gianniantonio Domina, Fabrizio Bartolucci, Gabriele Galasso, Lorenzo Peruzzi

**Affiliations:** 1Department of Architecture, University of Palermo, Viale delle Scienze, bldg. 14., I-90128 Palermo, Italy; 2Department of Agricultural, Food and Forest Sciences, University of Palermo, Viale delle Scienze, bldg. 4., I-90128 Palermo, Italy; 3Floristic Research Center of the Apennines, University of Camerino—Gran Sasso and Monti della Laga National Park, San Colombo, Barisciano, I-67021 L’Aquila, Italy; 4Section of Botany, Natural History Museum of Milan, Corso Venezia 55, I-20121 Milano, Italy; 5PLANTSEED Lab, Department of Biology, University of Pisa, Via Luca Ghini 13., I-56126 Pisa, Italy

**Keywords:** endemism, Mediterranean, nomenclature, taxonomy, typification

## Abstract

*Senecio squalidus* (Asteraceae) currently includes nine subspecies distributed in North and Central Europe and in the Mediterranean basin. Within this taxonomic aggregate, many species have been described, but research on their nomenclatural types is incomplete. A complete nomenclatural survey of 19 names belonging to this taxonomically critical group was carried out. Fourteen lectotypes are here designated. The nomenclatural analysis, complemented by field investigations in the type localities of the taxa described in the Central Mediterranean, allowed us to accept 10 species. Accordingly, we proposed here a new name and a new missing combination at a specific level: *S. aknoulensis* and *S. calabrus*.

## 1. Introduction

*Senecio squalidus* L. (Asteraceae) is a species recorded for North and Central Europe and the Mediterranean basin. Currently, nine subspecies are accepted as proposed by [1,2,3]. After [1], other contributions dealt with the nomenclature of this group [4,5,6], but many names still lack nomenclatural types.

In the study of this group, Sicily is a focal point because large part of the taxa have been described or reported from this island. For example, at Mount Etna, three taxa are found at different altitudinal belts. As reported by [7,8], *Senecio aethnensis* Jan ex DC. (≡*S. squalidus* subsp. *aethnensis* (Jan ex DC.) Greuter) grows on recent lava in the upper belt, and *S. siculus* All. (=*S. squalidus* subsp. *chrysanthemifolius* (Poir.) Greuter) in the basal belt. At intermediate elevations, these two taxa currently hybridize, forming a hybrid swarm. About 300 years ago, seeds of these hybrid populations were brought to England. At the end of the 18th century, after a century of cultivation in botanic gardens, stabilized derivatives of this old hybridization escaped and spread rapidly to many parts of the British Isles [9,10]. Accordingly, a homoploid hybridogenous species originated, different from the parentals but also from the original native hybrid [11,12,13,14,15,16,17]. Albeit homoploid hybrid speciation is relatively rare in Asteraceae [18], other examples of speciation due to hybridization are known from Europe, the Mediterranean, and the Americas in other genera of this family (*Bidens*, *Tragopogon*), but also in other lineages, such as *Oenothera* (Onagraceae), *Scabiosa* (Dipsacaceae), and *Vitis* (Vitaceae) [19,20,21,22].

Linnaeus [23] described *S. squalidus*, the oldest available name in this group, from cultivated plants possibly obtained by seeds received from Oxford [24], and thus this name refers to the hybridogenic species of the British Isles. On the contrary, the first name given to the hybrid occurring in the wild in Sicily (Mount Etna) is *S. ×glaber* Ucria [1,8].

*Senecio nebrodensis* L. is another species described from Sicily, Spain, and the Pyrenees. Alexander [1] lectotypified this name with a specimen from Spain. Recently it has been proposed as a nomen rejiciendum because it has been shown that the application of this name is ambiguous and based on a type that only partially corresponds to the original description [25]. The name proposed for these plants from Spain is *S. duriaei* J.Gay. Although there are different taxonomic opinions on this matter, the plants occurring in C and SE Europe were called *S. rupestris* Waldst. and Kit. (≡*S. squalidus* subsp. *rupestris* (Waldst. and Kit.) Greuter). The plants occurring elsewhere in Sicily and North Africa were referred by [1] as *S. squalidus* subsp. *aurasiacus* (Batt.) C. Alexander, and those from the high eastern end of the Riff Mountains in Morocco as *S. squalidus* subsp. *araneosus* (Emb. and Maire) C. Alexander. Finally, the populations growing in Sardinia and southern Calabria were classified as *S. squalidus* subsp. *sardous* Greuter and *S. squalidus* subsp. *calabrus* (Fiori) Peruzzi and Bernardo, respectively [6,26,27,28,29].

The morphological analysis of the original material traced for this study allowed us to better clarify the relationships among the taxa investigated, here considered at the species level, to define new synonyms, as well as to propose new missing names and combinations. This contribution is part of a series of investigations aimed at looking for the nomenclatural types of the plant taxa described in Italy [29,30,31,32].

## 2. Materials and Methods

Starting from the herbaria hosting the main collections of those authors involved in the *Senecio squalidus* group, we performed a survey of the original material available for the names involved. For the type of *S. squalidus*, we consulted the LINN herbarium (acronyms according to [33]); the herbaria of the National Museum (PR) and of the Charles University in Prague (PRC) for the names published by Carl Bořivoj Presl [34]; the Herbarium P for the collections by Jean Louis Marie Poiret and TO for the collections by Carlo Allioni [34]; the Herbarium of the Conservatoire et Jardin botaniques de la Ville de Genève (G) for the collections by Giorgio Jan, Jean Étienne Duby, and Alphonse Louis Pierre Pyramus de Candolle; and the herbarium of the Natural History Museum of Florence (FI) for the original material by Adriano Fiori. The herbarium by Bernardino da Ucria has been lost, so duplicates and other material were searched for in the main Italian and European herbaria BM, BOLO, BR, CAT, E, K, MA, MAF-POURRET, NAP, PAD, PAL, RO, W, WAG, and WU. The articles of the International Code of Nomenclature for algae, fungi, and plants (hereafter ICN) cited in the text follow [35]. In the proposed taxonomic treatment, species are arranged in alphabetical order. For each taxon, the accepted name is reported, as well as the synonyms, the publication details, type information, taxonomic notes, and distribution. The taxonomic revision is based on the morphology of the investigated specimens and supported by field investigations, carried out between 2019 and 2022, in type localities of the taxa described from Sicily, Calabria, and Sardinia. To facilitate identification, a dichotomous identification key and schematic drawings of capitula, upper leaves, and basal leaves arranged in the same order are presented.

## 3. Results

***Senecio aethnensis*** Jan ex DC., Prodr. 6: 345. 1838 [early January 1838].

≡*Senecio squalidus* var. *aethnensis* (Jan ex DC.) Fiori in Fiori, Béguinot and Paoletti, Fl. Italia 3(1): 213. 1903 [May 1903] ≡ *Senecio chrysanthemifolius* subsp. *aethnensis* (Jan ex DC.) Lambinon, Bull. Soc. Échange Pl. Vasc. Eur. Occid. Bassin Médit. 21: 63. 1986 [pubbl. after 19 Dec. 1986] ≡ *Senecio squalidus* subsp. *aethnensis* (Jan ex DC.) Greuter, Willdenowia 33(2): 248. 2003 [22 December 2003] ≡ *Jacobaea carnosa* C.Presl, Delic. Prag.: 93(–94). 1822. [July 1822] ≡ *Senecio gallicus* var. *aethnensis* Jan, Elench. Pl.: 14. 1827–1831 [31 March 1827–March 1831], nom. nud.—*Senecio carnosus* (C.Presl) C.Presl, Fl. Sicul.: XXVIII. 1826 [October 1826], non Lam. [1779], nom. illeg.

Ind. Loc.: “in acervis arenae vulcanicae regionis elatae montis Aethnae”.

Type (lectotype here designated):—[Sicily] *Senecio glaucus* Pr., Arena Aetna. Jul. ♃/*Senecio carnosus*, *Jacobaea carnosa* Presl, Sicilien, Juli, Juli-August (PR no. 4957, digital photo!).

The lectotype here selected is composed of a floral scape with eight fruiting capitula at different ripening stages and nine leaves. It includes the handwritten label by Carel B. Presl.

The specimen by Jan (G barcode G00471538 [digital photo!]), also mentioned in the protologue of *S. aethnensis*, applies to the same unit of diversity.

Taxonomic notes: This perennial species has lanceolate leaves with entire to dissected margin (Figure 1); capitula 7–13 mm in diameter; cypselae glabrous or puberulous, around 2.3–3.3 mm long.

Distribution: *Senecio aethnensis* is endemic to Mount Etna, where it grows on recent lava in the upper belt (1750–3050 m a.s.l.) [8] (Figure 2).

***Senecio aknoulensis*** Domina and Barone nom. and stat. nov.

≡*Senecio gallicus* subsp. *mauritanicus* var. *araneosus* Emb. and Maire, Mém. Soc. Sci. Nat. Maroc 17: 54. 1928 [31 December 1927 publ. after 31 July 1928] ≡ *Senecio squalidus* subsp. *araneosus* (Emb. and Maire) C. Alexander, Notes Roy. Bot. Gard. Edinburgh 37(3): 398. 1979 [December 1979].

Ind. Loc.: “In Callitrietis et Quercetis, solo margaceo, schistaceo: Aknoul; in monte Nador, Boured 800–1400 m”.

Type (lectotype (first-step) designated by [1] (p. 398)): [Morocco] In Atlante Rifano: in pascuis montis Nador 1400 m/19 June 1926, *R. Maire s.n*. MPU; (second-step designated here) MPU barcode MPU000224 [digital photo!]) (https://science.mnhn.fr/institution/um/collection/mpu/item/mpu000224, accessed on 1 August 2022).

Alexander [1] indicates the specimen “In Callitrietis et Quercetis, solo margaceo, schistaceo: Aknoul; in monte Nador, Boured 800–1400 m, 19 June 1926, Maire (MPU)” as an isotype and reports it as a poor specimen wherein it is hard to clearly distinguish the leaf characters. The type citation of [1] can be further narrowed to a single specimen by second-step lectotypification according to Art. 9.17 of ICN. In MPU, we found only a single specimen of this gathering (MPU000225), and another one (MPU000224) from the same locality, but with a different date.

Taxonomic notes: It is a perennial mountain species, characterized by abundant hairiness on the whole plant and lyrate leaves (Figure 1).

Distribution: *Senecio aknoulensis* is a narrow endemic species in the eastern end of the Riff Mts, N Morocco [1,36] (Figure 3).

***Senecio balansae*** Boiss. and Reut., Diagn. Pl. Orient. ser. 2, 3: 32(–33). 1856 [November–December 1856].

*≡Senecio nebrodensis* var. *aurasiacus* Batt. in Batt. and Trab., Fl. Algérie 1(3): 474. 1889 [November–December 1889] *≡ Senecio squalidus* subsp. *aurasiacus* (Batt.) C. Alexander, Notes Roy. Bot. Gard. Edinburgh 37(3): 397. 1979 [December 1979]—*Senecio nebrodensis* var. *balansae* (Boiss. and Reut.) Quézel and Santa, Nouv. Fl. Algérie 2: 961. 1963 [September 1963], comb. inval.

Ind. Loc.: “[Algerie] in declivitate boreali montis Gebel Tougour prov. Constantine cl. Balansa anno 1853”.

Type (lectotype here designated): [Algeria] *Senecio rupestris* Waldst. and Kit. Pente nord du Djebel-Tougour, prè[s] Batna, 20 July 1853, *B. Balansa* (P barcode P02685431 [digital photo!]) (https://science.mnhn.fr/institution/mnhn/collection/p/item/p02685431, accessed on 1 August 2022).

=*S. nebrodensis* var. *siculus* Fiori in Fiori, Béguinot and Paoletti, Fl. Italia 3(1): 212. 1903 [May 1903].

Ind. Loc.: “Rupi e boschi dalla reg. submont. alla mont. in Sic.—*S. nebrod*. Guss.”.

Type (lectotype here designated): [Sicily] *Senecio nebrodensis* L., Palermo: in montosis, May [18] 90, *H. Ross s. n*./*Senecio nebrodensis* L. var. *siculus* Nob., Adr. Fiori (FI barcode FI067113 [digital photo!]) (Figure 4).

Among the original material found in FI, there are some specimens prepared by other collectors and revised by Fiori. The indication “Nob.” in the label stands for “Nobis”, to indicate that the name was published by the same reviewer. We chose the specimen FI067113, collected by Hermann Ross, because it includes three well-grown plants with numerous leaves and flower heads at different developmental stages.

Taxonomic notes: It is an annual species previously confused with *S. siculus* [26] from which it differs in the leaves shallowly lobed and not pinnatifids (Figure 1).

Distribution: *Senecio balansae* occurs in Algeria (Lakhdaria, Aures, Constantine), Tunisia (Cedria, Dj. Zaghouan), and Sicily (Mount Gallo, Mount Pellegrino, Capo Zafferano, Mountains of Palermo, Busambra, Sicani Mountains, Madonie Mountains, Mountains of Trapani) (Figure 2 and Figure 3).

***Senecio calabrus*** (Fiori) Peruzzi comb. and stat. nov.

≡*Senecio nebrodensis* var. *calabrus* Fiori in Fiori, Béguinot and Paoletti, Fl. Italia 3(1): 212. 1903 [May 1903] *≡ Senecio squalidus* subsp. *calabrus* (Fiori) Peruzzi and Bernardo in Bernardo, Passalacqua and Peruzzi, Inform. Bot. Ital. 42(2): 531. 2010 [31 December 2010] (“*calabricus*”).

Ind. Loc.: “Monti di Cal. (Sila, Serra S. Bruno, M. Alto ecc.) e Sic. alle Madonie”.

Type (lectotype here designated): [Italy] *Senecio nebrodensis* L. var. *calabrus* Nob., Calabria: Sila a Torre Caprara, reg. mont. 18 June 1899, *A. Fiori* (FI barcode FI066024 [digital photo!]) (Figure 5).

Among the original material found in FI, there are some specimens prepared by other collectors and revised by Fiori. The indication “Nob.” in the label stands for “Nobis” to indicate that the name was published by the same collector. We chose the above-mentioned specimen, collected by Fiori himself, because it includes a plant with numerous flower heads at different developmental stages.

Taxonomic notes: It is an annual mountain species with narrowed obovate leaves divided into narrow and spaced laciniae (Figure 1). The lack of deep lobes in the base leaves allows for the distinction of this species from *S. rupestris* and *S. balansae*, whose ranges are in contact or, in part, overlap.

Distribution: *Senecio calabrus* in Calabria on the Sila Mts, Serra San Bruno, Aspromonte, and in Sicily in the higher Madonie Mountains ([28], field checks and specimens in FI) (Figure 2 and Figure 3).

***Senecio duriaei*** J.Gay ex DC., Prodr. 6: 350. 1838 [early January 1838].

—*Senecio duriaei* J.Gay, Ann. Sci. Nat., Bot. s. 2, 6(6): 346. 1836 [December 1836], nom. nud.

Type (lectotype designated by [25] (p. 1370): Durieu 306 (K barcode K000852081 [digital photo!], isolectotypes: BM barcode BM001042972 [digital photo!], G barcode G00471836 [digital photo!], P barcodes P03683332 [digital photo!], P03683326 [digital photo!], P03683313 [digital photo!], P03683308 [digital photo!]; image of the lectotype is available at http://apps.kew.org/herbcat/getImage.do?imageBarcode=K000852081, accessed on 1 August 2022).

=*Senecio nebrodensis* L., Sp. Pl., ed. 2: 1217. 1763 [August 1763], nom. rej. prop.

Ind. Loc.: “Habitat in Sicilia, Hispania, Pyrenaeis, Alstroemer”.

Type (lectotype designated by [1] (p. 394): Herb. Linnaeus no. 996.23 (LINN [digital photo!]) (https://linnean-online.org/10157, accessed on 1 August 2022).

In the protologue, this species is reported from Sicily, Spain, and the Pyrenees. This species is not known from the Pyrenees, and the populations from Spain and Sicily have been attributed to two different taxa [1]. Alexander [1] lectotypified this name with a specimen presumptively coming from southern Spain. Only recently has this name been proposed as a nomen rejiciendum [25] because it has been shown that its application is ambiguous and based on a type that only partially corresponds to the original description.

Taxonomic notes: This perennial species is characterized by basal lyrate leaves, lobed with crenate margins (Figure 1). It differs from *S. rupestris* by the presence of glandular hairs in the upper part of the stem.

Distribution: *Senecio duriaei* occurs from 1200 to 3000 m a.s.l. in central and southern Spain [37] (Figure 3).

***Senecio* ×*glaber*** Ucria, Nuov. Racc. Opusc. Aut. Sicil., 6: 255. 1793 (pro sp.).

≡*Senecio squalidus* subsp. *glaber* (Ucria) Nyman, Consp. Fl. Eur. 2: 357. 1879 [October 1879] ≡ *Senecio squalidus* var. *glaber* (Ucria) Fiori in Fiori, Béguinot and Paoletti, Fl. Italia 3(1): 213. 1903 [May 1903].—*Senecio glaber* Ucria, Arch. Bot. [Leipzig] 1(1): 70. 1796, isonym.

Ind. Loc.: [Etna].

Type (lectotype here designated): [Sicily] *Jacubaea Etnica, Chrysanthemi segetum, Lob. folio umbellifer*. Cupani Panphyton siculum pl. 139 [38] (pl. 364).

In the protologue of this name, after a short diagnosis, a synonym by Cupani is cited along with the reference to a plate of his *Panphyton Siculum*. No suitable original specimen by Ucria was found. The Herbarium Ucria has been lost and no specimen was found in PAL, FI, G, and in a small herbarium kept at “Società Siciliana per la Storia Patria di Palermo” attributed to Ucria [39]. For this reason, we here designate the plate 139 of the *Panphyton Siculum* by Cupani (Copy A, housed in the Central Library of Sicily Region) as the lectotype of the name. This iconography was known by scholars through the consultation of a dozen copies preserved in the main European libraries. These copies consisted of several engravings with different numbering [38]. Cupani’s work was published for the first time in 2003 by collating the main existing copies [38].

≡*Jacobaea incisa* C.Presl, Delic. Prag.: 94(–95). 1822 [July 1822] *≡ Senecio incisus* (C.Presl) C.Presl, Fl. Sicul.: XXVIII. 1826 [October 1826], non Thunb. [1800], nom. illeg. *≡ Senecio aethnensis* var. *incisus* (C.Presl) DC., Prodr. 6: 345. 1838 [early January 1838] *≡ Senecio squalidus* var. *incisus* (C.Presl) Guss., Fl. Sicul. Syn. 2(1): 475. 1843–1844 [August 1843–June 1844].

Ind. Loc.: “Hab. in acervis vulcanis regionis apertae montis Aetnae”.

Type (lectotype here designated): [Sicily] *Jacubaea Etnica, Chrysanthemi segetum, Lob. folio umbellifer*. Cupani Panphyton siculum pl. 139 [38] (pl. 364).

No original specimen has been found in the investigated herbaria. We here designate as the lectotype of the name the iconography included in the copy by Bonanno reported in the protologue. The plate 167 of Bonanno’s copy corresponds to the plate 139 of Cupani’s copy and to the plate 364 of the 2003 edition [38]. After lectotypifications, *Senecio ×glaber* Ucria and *Jacobaea incisa* C.Presl are homotypic.

Taxonomic notes: *Senecio ×glaber* is a perennial taxon with entire, lanceolate, rarely bipinnatifid leaves; capitula of 7–10 mm in diameter (Figure 1); cypselae glabrous or puberulous, around 2.3–2.6 mm long. The main differences between *S. ×glaber* and *S. squalidus* lie in the leaf shape (pinnatifid in *S. squalidus* vs. entire and dentate in *S. ×glaber*) and in the size of capitula (20–22 mm in diameter in *S. squalidus* vs. 7–10 mm in *S. ×glaber*).

Distribution: This morphologically variable hybrid occurs in Mount Etna in the contact zone between *S. aethnensis* and *S. siculus*, at altitudes between 1300 and 1900 m a.s.l. [8] (Figure 2).

***Senecio rupestris*** Waldst. and Kit., Descr. Icon. Pl. Hung. 2: 136(–137, pl. 128). 1803–1805.

*≡Senecio nebrodensis* subsp. *rupestris* (Waldst. and Kit.) Hayek, Repert. Spec. Nov. Regni Veg. Beih. 30(2): 682. 1931 [15 February 1931] *≡ Senecio squalidus* subsp. *rupestris* (Waldst. and Kit.) Greuter, Willdenowia 35(2): 238. 2005 [21 December 2005] *≡ Senecio squalidus* var. *rupestris* (Waldst. and Kit.) P.D.Sell in P.D.Sell and G.Murrell, Fl. Gr. Brit. Ireland 4: 556. 2006 [1 April 2006]—*Senecio squalidus* subsp. *rupestris* (Waldst. and Kit.) P.D.Sell in P.D.Sell and G.Murrell, Fl. Gr. Brit. Ireland 4: 556. 2006 [1 April 2006], isonym.

Ind. Loc.: “Crescit in rupibus calcareis Croatiae ad Koreniczam: velut in monte Merzin, in valle Villena draga, atque in ipsis alpibus Plissivicza and Velebich. Eudem legimus infra Rézbányam in arena rivi, sine dubio e montibus per aquam delatum”.

Type (lectotype designated by [40] (p. 220): PR155832/788 a, b. Plate 12.

=*Senecio laciniatus* Bertol., J. Bot. Agric. 2(2): 76. 1813 [August 1813] *≡ Senecio nebrodensis* var. *laciniatus* (Bertol.) Batt. in Batt. and Trab., Fl. Algérie 1(3): 474. 1889 [November–December 1889].

Ind. Loc.: none.

Type (neotype designated by [5] (p. 50): *Jacobaea fol. minus dissecti amplo flori*, s.d., *G. Monti s.n*. (BOLO ex Herb. Monti [digital photo!]).

Taxonomic notes: This perennial species shows lanceolate basal leaves, deeply laciniate with toothed margins (Figure 1). *S. rupestris*, compared to the other species of the group, shows a relative morphological homogeneity throughout its range.

Distribution: *Senecio rupestris* is a widely distributed species, found in central Europe, peninsular Italy, and the Balkan peninsula (Figure 3).

***Senecio sardous*** (Greuter) Arrigoni, Parlatorea 9: 94. 2007 [October 2007].

≡*Senecio leucanthemifolius* var. *nemoralis* Gennari, Sp. Fl. Sarda: 30. 1867 ≡ *S. nebrodensis* var. *sardous* Fiori in Fiori, Béguinot and Paoletti, Fl. Italia 3(1): 212. 1903 [May 1903], nom. illeg. ≡ *Senecio squalidus* subsp. *sardous* Greuter, Willdenowia 35(2): 238. 2005 [23 December 2005].

Ind. Loc.: “Montagne di Osidda, e di Capoterra”.

Type (lectotype here designated): [Sardinia] *Senecio leucanthemifolius* var. *nemoralis*, Capoterra, May 1862, [*Gennari*] (TO no. 4025 [digital photo!]).

The type indicated by [6] (p. 94) for “*S. nebrodensis* var. *sardous* Fiori” [Giovannibono (S. Vito-Sarrabus), in sylvis, 23 April 1872, *S. Sommier s.n*. (FI)], cannot be accepted since *S. nebrodensis* var. *sardous* is a superfluous and illegitimate name (Arts. 52.1–2 of the ICN) homotypic with *S. leucanthemifolius* var. *nemoralis* Gennari, described in 1867. Hence, the name by Greuter is not a new combination but a replaced name at a different rank (Art. 58.1 of the ICN), as already evidenced by [29].

Taxonomic notes: It is an annual species, which differs from *S. balansae* in having non-lyrate basal and upper leaves.

Distribution: *Senecio sardous* is endemic to Southern and Central Sardinia, where it grows from sea level to 1200 m a.s.l. [6] (Figure 3).

***Senecio siculus*** All., Auct. Syn. Meth. Stirp. Hort. Regii Taur.: 18. 1773 [30 September 1773].

Ind. Loc.: [Sicily].

Type (lectotype here designated): [Sicily] *Senecio siculus, chrysanthemifol*. Cyr. (TO ex Herb. Allioni [digital photo!]) (Figure 6).

Only the specimen here designated as lectotype was found in the Herbarium Allioni in TO. It consists of a single plant, without roots, with three flower scapes rich in deeply laciniate leaves and about 20 flower heads at different degrees of development. This specimen corresponds well to Boccone’s iconography [41] (pl. 36 at p. 67) related to the diagnosis by Boccone himself [41] (p. 66) and reproduced in Allioni’s protologue. It was chosen here as the lectotype of the name because it is more complete.

=*Senecio chrysanthemifolius* Poir., Encycl. [J. Lamarck and al.] 7: 96. 1806 [6 July 1806] ≡ *Senecio squalidus* var. *chrysanthemifolius* (Poir.) Guss., Fl. Sicul. Syn. 2(1): 476. 1843–1844 [August 1843–June 1844] ≡ *Senecio squalidus* subsp. *chrysanthemifolius* (Poir.) Greuter, Willdenowia 33(2): 248. 2003 [22 December 2003].

Ind. Loc.: “Sicile”.

Type (lectotype here designated): [Sicily] *Jacobea, chrysanthemi facie* Boccone, Icon. and Descr. Rar. Pl. Sic., Mel., Gall., Ita.: 67. pl. 36. 1674.

The Boccone’s iconography mentioned in the protologue is designated as the lectotype of the name. It represents a well-grown perennial plant with a dozen flower heads at different levels of development.

*=Senecio chrysanthemifolius* var. *microglossus* DC., Prodr. 6: 345. 1838 [early January 1838] *= Senecio squalidus* var. *microglossus* (DC.) Guss., Fl. Sicul. Syn. 2: 476. 1844 [June 1844] *= Senecio squalidus* subsp. *microglossus* (DC.) Arcang., Comp. Fl. Ital.: 345. 1882 [January–April 1882].

Ind. Loc.: “circa Cataniam legit cl. Duby (v. s.)”.

Type (lectotype here designated): [Sicily] Catania, 1832, *M. Duby s.n*. (Herb. G-DC barcode G00471562 [digital photo!]) (https://www.ville-ge.ch/musinfo/bd/cjb/chg/adetail.php?id=328846&base=img&lang=en, accessed on 1 August 2022).

In G-DC, only one specimen was found, but in the impossibility of knowing whether there were more duplicates, this specimen is prudently designated here as a lectotype. *Senecio chrysanthemifolius* var. *microglossus* was distinguished for having small, revolute ligules, slightly exceeding the flower head. This taxonomic character is not constant in nature, and the low-altitude populations of Mount Etna all refer to *S. siculus*.

Taxonomic notes: It is a perennial species with bipinnatifid, deeply lobed leaves; capitula 6.5–9.0 mm in diameter (Figure 1); cypselae hairy, around 2.1–2.2 mm long.

Distribution: From the studied herbarium specimens housed in FI, P, PAL, and the field investigations, we can state that *Senecio siculus* is endemic to Sicily, southern Italy, and northern Tunisia. It occurs in Mount Etna in the basal belt, up to about 1500 m a.s.l., and grows also in northern and southern Sicily, the Aeolian islands, Calabria (S Italy), and Cape Bon (N Tunisia) (Figure 2 and Figure 3).

***Senecio squalidus*** L., Sp. Pl. 2: 869. 1753 [1 May 1753].

≡*Jacobaea squalida* (L.) C.A.Mey., Verz. Pfl. Casp. Meer.: 81. 1831 [November–December 1831].

Ind. Loc.: “Habitat in Europa Australi”.

Type (lectotype designated by [4] (p. 366): Herb. Linn. No. 996.33 (LINN [digital photo!]) (http://linnean-online.org/10167/, accessed on 1 August 2022).

Taxonomic notes: It is a perennial stabilized hybrid species that originated in England from seeds of *S. ×glaber* Ucria collected on Mount Etna [11]. This species has elliptic to oblong basal leaves, pinnatifid to pinnatipartite, rarely entire and dentate, middle and upper stem leaves pinnatifid to pinnatipartite, less frequently entire (Figure 1).

Distribution: Starting from the UK, *S. squalidus* became later naturalized in France, northern Europe, the USA, and Canada [42] (Figure 3). The wide distribution reported throughout Europe (especially in Mediterranean countries), Algeria, and Morocco [42,43] refers to the whole taxonomic group.

### Identification Key to the Species of Senecio Squalidus Group


**1.**
Basal leaves pinnatifid to pinnatipartite
**2**

**1.**
Basal leaves entire to shallowly lobed
**7**

**2.**
Leaf lobes narrow and long (length/width ratio > 3), with acute tips
**3**

**2.**
Leaf lobes wide and short (length/width ratio ≤ 3), with rounded tips
**5**

**3.**
Whole plant arachnoid, basal leaves widened at the apex
**
*S. aknoulensis*
**

**3.**
Whole plant glabrous to sparsely arachnoid, basal leaves not or slightly widened at the apex
**4**

**4.**
Leaves coarsely divided, lobes 3–5 mm wide or more
**
*S. squalidus*
**

**4.**
Leaves finely divided, lobes 1–3 mm wide
**
*S. siculus*
**

**5.**
Basal leaves lyrate, with apex at least twice as wide as the lateral lobes
**
*S. balansae*
**

**5.**
Basal leaves not lyrate, with narrow apex, slightly wider than the lateral lobes
**6**

**6.**
Plant glandular; leaves lobed, with crenate margin
**
*S. duriaei*
**

**6.**
Plant eglandular; leaves deeply laciniate, with toothed margin
**
*S. rupestris*
**

**7.**
Leaves thick and rigid
**8**

**7.**
Leaves thin and soft
**9**

**8.**
Leaves with short and acute lobes
**
*S. ×glaber*
**

**8.**
Leaves with entire to dissected margins
**
*S. aethnensis*
**

**9.**
Basal leaves not or slightly widened at the apex
**
*S. calabrus*
**

**9.**
Basal leaves widened at the apex
**
*S. sardous*
**


## 4. Conclusions

On the whole, 19 names belonging to the *Senecio squalidus* group were considered. For five of them (*S. duriaei*, *S. laciniatus*, *S. nebrodensis, S. rupestris*, and *S. squalidus*), types were already designated by previous scholars. For *Senecio gallicus* subsp. *mauritanicus* var. *araneosus* Emb. and Maire, a second-step lectotype was designated. For 10 names (*Jacobaea carnosa*, *Senecio aethnensis*, *S. balansae, S. chrysanthemifolius* var. *microglossus, S. leucanthemifolius* var. *nemoralis*, *S. nebrodensis* var. *aurasiacus*, *S. nebrodensis* var. *calabrus, S. nebrodensis* var. *siculus, S. sardous*, and *S. siculus*), lectotypes are here designated on herbarium specimens; for three (*Jacobaea incisa*, *Senecio chrysanthemifolius*, and *S. ×glaber*), lectotypes are here designated on iconographies.

We agree with the proposal [25] of rejecting the name *S. nebrodensis* because it has often been treated as a nomen confusum. Furthermore, in the European Herbaria, the Spanish specimens are either casually identified as *S. nebrodensis* or *S. duriaei*.

It is possible to discriminate the taxa through the morphology of the basal leaves. This character is also easily observed in the photos of herbarium specimens. Some taxa are very recognizable by their diffuse hairiness (*S. aknoulensis*), by the presence of glandular hairs in the upper part of the stem (*S. duriaei*), or by their complete absence of hairs (*S. aethnensis*). The size of cypselae is also a relevant and important discriminatory feature [1], but it is not always observable in herbarium specimens.

The duration of the life cycle clearly distinguishes two groups: annuals (*S. balansae*, *S. calabrus*, and *S. sardous*) and short-lived perennials (*S. aethnensis*, *S. aknoulensis*, *S. duriaei*, *Senecio ×glaber*, *S. rupestris*, *S. siculus*, and S. *squalidus*).

*Senecio squalidus* is morphologically very variable, and *S. ×glaber* shows a range of morphological variations from one parental taxon to the other. On the contrary, *S. rupestris* is morphologically very homogeneous throughout its range.

The taxonomic relationships among the taxa investigated are not clear yet, and in several cases, there are geographical and/or ecological overlaps. The previous taxonomic accounts [1,2,3], which have seen all taxa at the subspecific level within *S. squalidus*, is not adequate to explain the variability that has emerged so far. Compared to previous taxonomic treatments, we deem more appropriate to treat these taxa at the species level, given their morphological diagnosability but unclear systematic and phylogenetic relationships. Accordingly, a new name (*S. aknoulensis*) and a new missing combination at a specific level (*S. calabrus*) are here proposed.

Two new synonymies are proposed here: *S. chrysanthemifolius*, which was previously accepted at different ranks [1,2,3,44], and *S. squalidus* subsp. *microglossus*, also accepted [2,3,44]. They are both regarded as heterotypic synonyms of *S. siculus*.

Consequently, the current taxonomic treatment accepts 10 species: *S. squalidus*, a hybridogenous species known only as a naturalized alien in England, northern France, the USA, and Canada [42,45]; *S. ×glaber* (*S. aethnensis* × *S. siculus*), a hybrid endemic to Mount Etna between 1300 and 1900 m a.s.l.; *S. aethnensis* endemic to the upper belt of Mount Etna; *S. siculus* endemic to Sicily, the extreme southern peninsular Italy, and Tunisia, from the sea level up to 1500 m a.s.l.; *S. duriaei* endemic to Spanish mountains; the south-eastern European *S. rupestris*; *S. balansae* endemic to Sicily, Tunisia, and Algeria; *S. calabrus* endemic to the mountains of Calabria and Sicily; *S. sardous* endemic to Sardinia; and *S. aknoulensis* endemic to the mountains of N. Morocco. From a biogeographical point of view, Sicily is the territory that hosts the largest number of taxa within this group. This could be explained by adaptive radiation [46,47], a phenomenon that is however still poorly understood in Mediterranean plants [48]. On the basis of the data available in the literature, *S. squalidus* and all the species native to Central Mediterranean are seemingly diploid with 2*n* = 20 chromosomes [1,49,50,51,52,53,54,55,56,57], while *S. duriaei* is reported as tetraploid [56,57]. A molecular phylogeny should be carried out to properly address the systematic relationships among these morphospecies.

## Figures and Tables

**Figure 1 plants-11-02597-f001:**
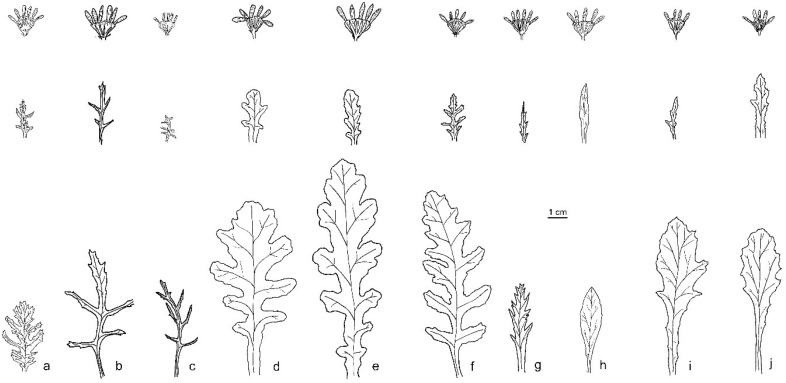
Schematic drawings of capitula, upper leaves, and basal leaves of the studied taxa of *Senecio* from the original material or specimens collected in the type localities: (**a**) *S. aknoulensis* (MPU000224); (**b**) *S. squalidus* (Herb. Linn. No. 996.33, LINN); (**c**) *S. siculus* (*Senecio siculus*, *chrysanthemifol*. Cyr. (TO ex Herb. Allioni); (**d**) *S. balansae* (P02685431); (**e**) *S. duriaei* (K000852081); (**f**) *S. rupestris* (L3657828); (**g**) *S. ×glaber* (CAT23508); (**h**) *S. aethnensis* (G00471538); (**i**) *S. calabrus* (FI066024); (**j**) *S. sardous* (TO4025). Drawings by G. Domina.

**Figure 2 plants-11-02597-f002:**
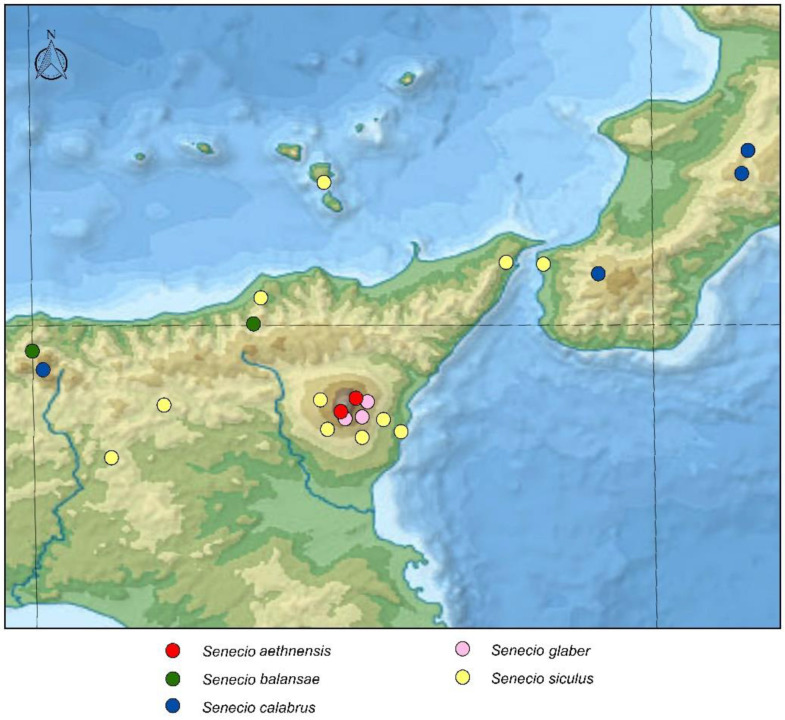
Detailed distribution map of the five species occurring across eastern Sicily and southern Calabria according to the studied herbarium specimens.

**Figure 3 plants-11-02597-f003:**
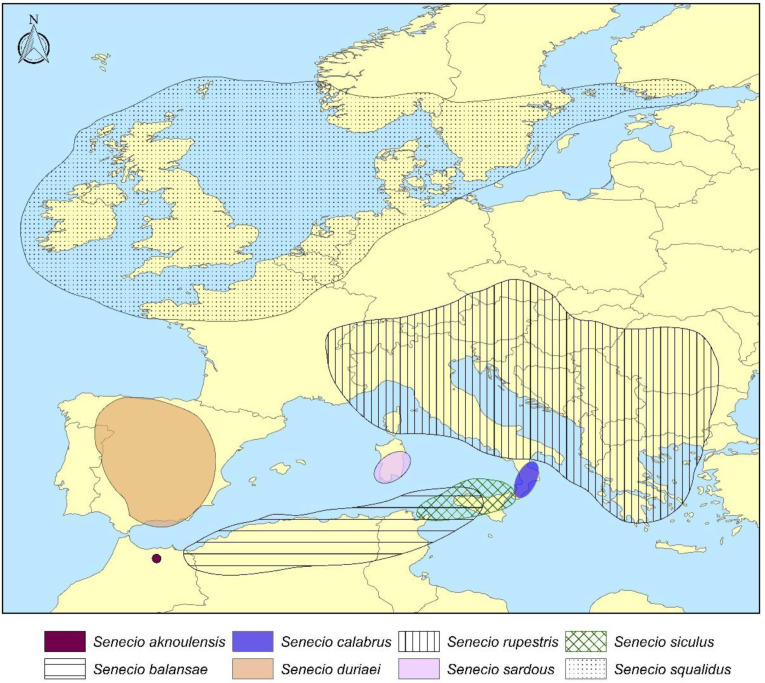
Overview distribution map concerning 8 out of the 10 species of the *Senecio squalidus* group.

**Figure 4 plants-11-02597-f004:**
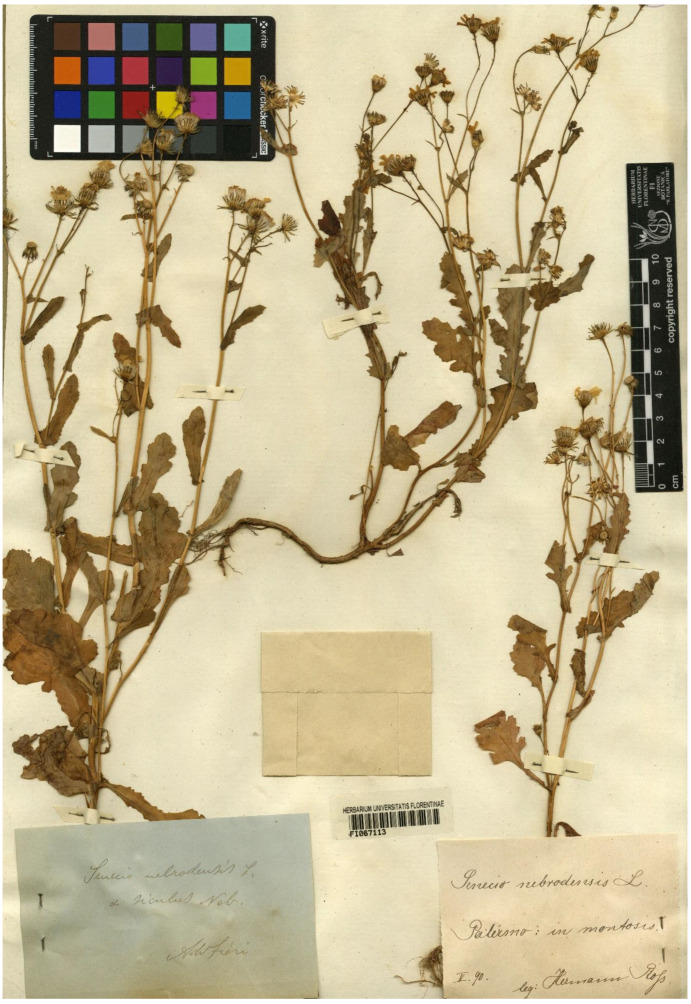
The specimen FI067113 collected by Hermann Ross and revised by Adriano Fiori, here designated as lectotype of the name *Senecio nebrodensis* var. *siculus* Fiori (photo reproduced with permission).

**Figure 5 plants-11-02597-f005:**
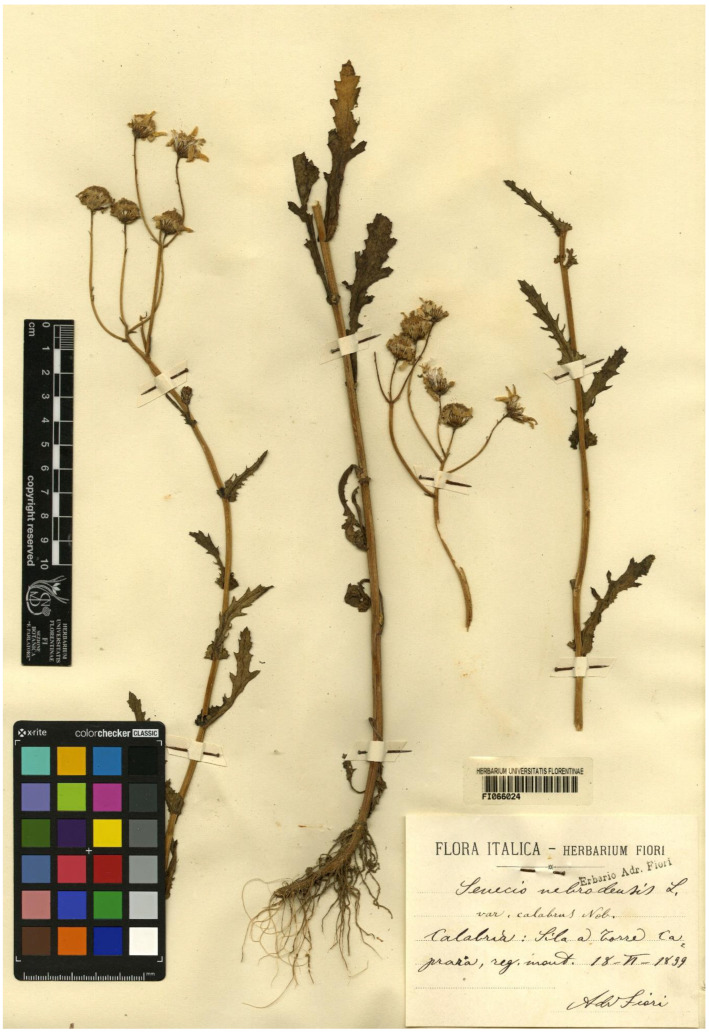
The specimen FI066024 collected by Adriano Fiori and here designated as lectotype of the name *Senecio nebrodensis* var. *calabrus* Fiori (photo reproduced with permission).

**Figure 6 plants-11-02597-f006:**
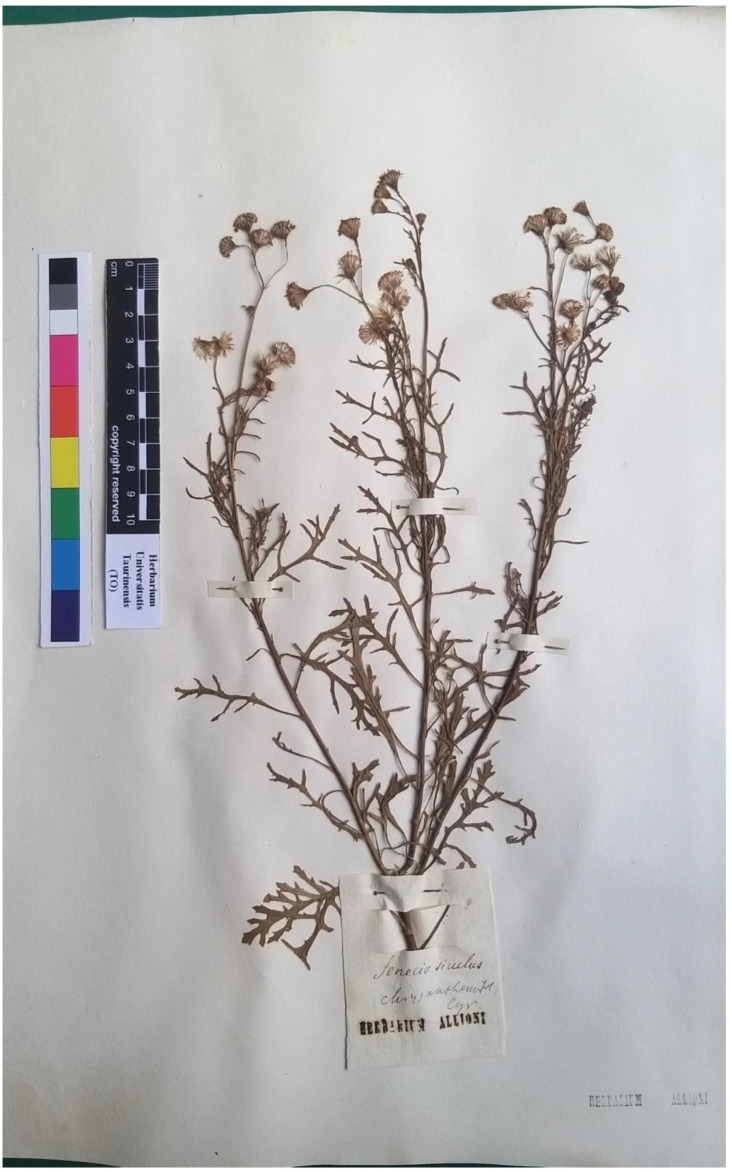
The specimen conserved in the Herbarium Allioni, TO, here designated as lectotype of the name *Senecio siculus* All. (photo reproduced with permission).

## Data Availability

The data presented in the current study are available within the article.

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
