# Peer review of "A Nomenclatural and Taxonomic Revision of the Senecio squalidus Group (Asteraceae)"

_plants, 2022, doi:10.3390/plants11192597_

Round 1
Reviewer 1 Report
This is a nice nomenclatural study on a taxonomically difficult group. Although the paper is purely technical, it was interesting to read the paper. I trust the data quality and the authors' competence and believe that this would be a valuable addition to the taxonomy of Senecio.
In spite of the merits, this submission has its apparent shortcomings which require attention. First of all, the authors presented this study as purely nomenclatural and get duly concentrated on its nomenclatural details - but it is not only a nomenclatural inventory but also a taxonomic evaluation / overview of this very difficult group. This is the right thing to do, of course - nomenclature has sense only together with taxonomy - but the authors forgot to include the taxonomic background. I see that the authors made a taxonomic revision and introduced some new species-level taxa - but what is their taxonomic concept? Why no discussion is provided about the diagnostic characters in this group and their significance and possible variability? This taxonomic background must be provided in order to explain the new system of the group proposed by the authors.
For this reason I suggest to change the title of this paper, saying straightforward that it is a taxonomic and nomenclatural revision of the group, rather than just "novelties".
To present the taxonomy even better, I suggest to provide an overview map with approximate outlines of species distributions. Please make a separate statement about distribution for each accepted taxon (now distributions are hidden in Notes).
Minor note: "specimen" is more correct an English word for "exsiccatum" used by the authors.
Author Response
Dear reviewer,
We have applied all the requested corrections and we believe that now the text has gained in clarity and errors have been removed.
In addition to the suggestions of the reviewers we have proposed a new name at specific rank (Senecio aknoulensis) for Senecio squalidus subsp. araneosus because Senecio araneosus is already in use for an Asiatic species. We verified that the original Spelling of S. duriaei is correct, so no need to correct it.
In detail:
- The taxonomic concept adopted was better explained in the paragraph Conclusion
- Diagnostic characters were better commented in the paragraph Conclusion
- The title of the article was changed
- An overview map and a detail map for eastern Sicily were added (Figures 5 and 6)
- Distribution is now presented for each accepted taxon in a dedicated paragraph.
- The word "exiccatum" was replaced with "specimen"
Best Regards
Gianniantonio Domina
Reviewer 2 Report
- The paper is quite interesting and the authors give light to a needed clarification of the taxonomy of this complex group of plants. The inclusion of the identification key together with the figure of the leaves are quite helpful for further identification of these plants. However, and after reading the paper, I have found some minor flaws which are required to be resolved.
- p 44-45. I recommend to add the name of the families between brackets.
- p 50. The Pyrenees are also part of Spain. Therefore, and related to the current mention of Spain, I recommend to add which geographic part of this country you are referring to.
- Senecio araneosus: Alexander reported an isotype ketp at MPU, and he did not specifically mention the number of the voucher. Review the ICN, and maybe this might be also treated as a new lectotype or as a second-step lectotype.
- Senecio balansae: This name seems not to be typified, and the ‘in loc’ mention is also missed for this name. Review it. Regarding Fl067113, there are three pieces of plants, and two of them appears together with a different label, respectively. Do they correspond to different gatherings? Are the remaining potential lectotypes similar to this one? I have a concern about the possibility to contain different gatherings made at different times, that a specimen is a gathering or part of a gathering made at one time (Art. 8.2 of the ICN).
- Senecio calabrus, Senecio balansae: What is Nob? This name/abbreviation appears in the labels and no mention or explanation is given.
- Senecio durieui: In Alexander’ paper, there is not direct indication of the typification of the name, and the mentioned page corresponds to S. squalidus subsp. squalidus. In addition, the cited Linn voucher is 996.33 and the authors mentions 996.23. Review it.
Author Response
Dear Reviewer,
We have applied all the requested corrections and we believe that now the text has gained in clarity and errors have been removed.
In addition to the suggestions of the reviewers we have proposed a new name at specific rank (Senecio aknoulensis) for Senecio squalidus subsp. araneosus because Senecio araneosus is already in use for an Asiatic species. We verified that the original Speccling of S. duriaei is correct, so no need to correct it.
In detail:
- p. 44-45. Added the name of the families
-p. 50. the sentence about Pyrenees was corrected.
- Senecio gallicus subsp. mauritanicus var. araneosus: Was added a discussion about the validity of the Lectotypification by Alexander. No need to specify the specimen number to be valid his designation.
- Senecio balansae: We confirm that the name Senecio balansae is here typified and the indication of the locus is given (lines 141-149). The specimen FI067113 (in Figure 1) includes 3 parts of plants presuntively from the same gathering. The two labels are by Hermann Ross (the collector) dated 1890 and by Adriano Fiori (who reviewed the specimen at the beginning of the XX Century). Now we explained it better in the text.
- Was explained in the text the meaning of "Nob." appearing in a couple of herbarium labels.
- The specimen LINN 996.23 is the Lectotype of Senecio nebrodensis we have corrected tha page of Alexander not 396 but 394.
Best Regards
Gianniantonio Domina
Reviewer 3 Report
Reviewer Report
Overview
The manuscript deals with complete nomenclatural survey of a large number of taxa in Solidago squalidus group. The authors have made a very thorough search for the original materials in many herbaria and additionally they carried out field investigations in the type localities. The subject of this research is very important considering that nomenclatural studies are essential for the taxonomic assessments.
General comments
The Title of the publication is appropriate. The Abstract is well built and provide a sufficiently detailed summary of the research and outputs. The Introduction is informative, presents the problem undertaken and define the purpose of the work. The methodology and approach to the topic is good. The Results and Conclusions are explicit. Figures are appropriate and of good quality.
However, there are still some minor editing required and some errors that should be considered before publication.
Specific comments
1. To the Abstract:
· - The sentence "Thirteen lectotypes and are here designated." should be revised as it seems to me that some word is missing here ("and are").
2. To the Introduction:
· - In page 1, line 28: "9" should be corrected to "nine".
[Comment – please, be consistent in the use of numerals or spelled-out numbers. Generally, words are used for numbers one through nine, and numerals for all other numbers - 10 and over 10. This is not always followed in the current paper, e.g. page 1, line 28 ("9"); page 3, lines 103-104 ("8" and "nine", respectively). Whatever you choose, be consistent].
· - Page 2, line 49: It seems to me that it would be better if there is a reference (IF available) included at the end of the sentence "On the contrary, the first name given 48 to the hybrid occurring wild in Sicily (Mount Etna) is S. ×glaber Ucria". However, if such reference is included, attention must be paid to all following reference numbers.
3. To the Results:
· - Page 3, line 134-135: "Senecio rupestris Waldst. & Kit. Route nord 134 du Djebel Tougour, pré Batna, 20 July 1853, B. Balansa" – my advice to the authors is to check carefully the source (the photo with the hand-written label and especially the link with the verbatim locality) and to make the necessary corrections.
· - Page 5, line 159: "Bernardo, N.G.Passal. Peruzzi," – this needs correction to be properly cited.
· - Page 5, line 162-164: Is there a link to the digital photo of this specimen (FI barcode FI066024)?
· - Page 5, lines 175 and 183; Page 6, line 248 and 261: concerning the citation of authors who designated types – the authors should be consistent and follow the model applied in the manuscript to the other designations [i.e. "... (lectotype designated by [25] (p. 1370): ..."].
· - Page 7, line 267: "This species endemic to …" should be corrected to "This species is endemic to …".
· - Page 7, line 279: "… related to the diagnosis by Boccone himself [42] (p. 66) reproducted in the Allioni’s …" – and related?
· - Page 7, line 288: "pag." could be omitted.
· - Page 7, line 307: The two parts in this sentence could be split and become two separate sentences.
· - Page 9, line 328: "(Hind & King 2022)" should be corrected to "[44]".
· - Page 10, line 356: "b) Senecio squalidus" should be corrected to "b) S. squalidus".
· - Page 10, line 384: "(S. aethnensis × S. siculus) hybrid endemic to …" should be corrected to "(S. aethnensis × S. siculus) a hybrid endemic …"
4. To the Author contributions:
· - A check of some author initials is needed.
5. To the References:
· - I am not convinced that it is necessary to add the reference [35] in Materials and Methods; Stafleu & Cowan [34] is enough. But this is authors' decision.
· - The references should be checked and corrected at some places noted by me in the PDF-file.
Other (minor) comments about the use of commas, full stops, quotation marks, semicolons, blank spaces, italicisation of Latin names, and orthographic mistakes are included in the PDF-file. Please, see them and make the necessary corrections.
Summing up, the subject of the paper falls within the general scope of the journal. The overall impression of the manuscript is good. It is well written, the research is well organized and executed, and results are clearly presented. The construction of the chapters is proper and logical in accordance with the task.
Decision: I suggest acceptance for publication after a few minor revisions.

Author Response
Dear Reviewer,
thanks for your work on this text. We have applied all the requested corrections and we believe that now the text has gained in clarity and errors have been removed.
In addition to the suggestions of the reviewers we have proposed a new name at specific rank (Senecio aknoulensis) for Senecio squalidus subsp. araneosus because Senecio araneosus is already in use for an Asiatic species. We verified that the original Speccling of S. duriaei is correct, so no need to correct it.
We include both the File with the track-changes on to have an idea of the work done and the clean version with the figures updates.
We suggest to work on this version for further data processing.
In this moment the figures are embendded in th text. High resolution figures are available.
Below the replies in detail to the single reviewers:
In detail:
- We corrected the sentence in the abstract "Thirteen lectotypes are here designated."
- We corrected the use of numerals or spelled-out numbers
- We added a reference to the sentence "On the contrary, the first name given to the hybrid occurring wild in Sicily (Mount Etna) is S. ×glaber Ucria" this datum can be retrived from the names reported in Alexander (1979) and Fici & Lo presti (2003)
- A figure with the specimen FI066024 was added
- citations of authors who designated types were arranged according to the instructions for the authors of the journal.
- The check of the author initials was done and the mistakes corrected.
- Reference 35 was deleted and the following ones were re-numbered
Best Regards
Gianniantonio Domina
Round 2
Reviewer 1 Report
The text has been considerably improved and major problems fixed.
Regarding particular points, I doubt one typification as described by the authors. "var. araneosus" was lectotypified with the "isotype at MPU"; this designation concerns a certain gathering but the holotype/lectotype specimen was not indicated in that type designation (isotype at MPU but holotype at some uncertain place). For this reason, the lectotype cannot be treated as designated by that statement, and a new (second-step) lectotypification is needed. A list of articles cited by the authors for this case is mostly irrelevant to the case.
The ascription of Senecio aethnensis to "Jan ex DC." seems to be erroneous because Jan proposed a variety of S. gallicus but not the species as accepted by Candolle.
Line 300: "nemorensis" is an error for "leucanthemifolius".
Please proofread the text very thoroughly for the language.
Author Response
Dear Editor, dear reviewer,
- For the name Senecio gallicus subsp. mauritanicus var. araneosus Emb. & Maire a secod-step lectotype was designated according to the reviewer's suggestion.
The indication "Jan ex DC." is correct. It does not come from the name proposed by Jan but from the protologue of Senecio aethnensis in which De Candole says that he is referring to "Senecio aetnensis" of Jan pl. exs. 1831. So the name Senecio aethenensis is published by DC. that refers the name to Jan. "Jan ex DC." precisely.
https://www.biodiversitylibrary.org/item/7155#page/350/mode/1up
https://www.ville-ge.ch/musinfo/bd/cjb/chg/adetail.php?id=328771&base=img&lang=en
Corrected "S. nebrodensis var. nemoralis" with "S. leucanthemifolius var. nemoralis" in line 300.
The text was proofreaded for the language.